# Unscrambling disease progression at scale: fast inference of event permutations with optimal transport

**Peter A. Wijeratne**
Sussex AI Centre
Department of Informatics
University of Sussex
Brighton, BN1 9RH
United Kingdom
`p.wijeratne@sussex.ac.uk`

**Daniel C. Alexander**
Hawkes Institute
Department of Computer Science
University College London
London, WC1E 6BT
United Kingdom
`d.alexander@ucl.ac.uk`

## Abstract

Disease progression models infer group-level temporal trajectories of change in patients' features as a chronic degenerative condition plays out. They provide unique insight into disease biology and staging systems with individual-level clinical utility. Discrete models consider disease progression as a latent permutation of events, where each event corresponds to a feature becoming measurably abnormal. However, permutation inference using traditional maximum likelihood approaches becomes prohibitive due to combinatoric explosion, severely limiting model dimensionality and utility. Here we leverage ideas from optimal transport to model disease progression as a latent permutation matrix of events belonging to the Birkhoff polytope, facilitating fast inference via optimisation of the variational lower bound. This enables a factor of 1000 times faster inference than the current state of the art and, correspondingly, supports models with several orders of magnitude more features than the current state of the art can consider. Experiments demonstrate the increase in speed, accuracy and robustness to noise in simulation. Further experiments with real-world imaging data from two separate datasets, one from Alzheimer's disease patients, the other age-related macular degeneration, showcase, for the first time, pixel-level disease progression events in the brain and eye, respectively. Our method is low compute, interpretable and applicable to any progressive condition and data modality, giving it broad potential clinical utility.

## 1 Introduction

The main aim of disease progression modelling is to learn a hidden underlying disease trajectory from 'snapshots' (sets of observations at a single time) of individuals at hidden points along the trajectory. The classical approach is to treat the problem dynamically, using either discrete [1–9] or continuous [10–18] models with latent variables to describe the hidden disease stage or time. An abundance of such models have been proposed (see [19] for a comprehensive review) and have found extensive success in providing unique interpretability and utility across a wide range of progressive diseases, including Alzheimer's disease (AD) [1, 20, 5, 12, 21, 22], Huntington's disease [23–26], multiple sclerosis [27, 28], Parkinson's disease [29], prion disease [30], amyotrophic lateral sclerosis [31], and chronic obstructive pulmonary disorder [32].

However all previous approaches make a compromise: they are either i) interpretable in feature space but sacrifice computational tractability [33, 1, 34, 2, 5, 13, 35, 6, 36–39]; or ii) are made computationally tractable by encoding to a latent space but sacrifice direct interpretability [40, 41]. Models of type (i) often require preprocessing or dimensionality reduction to extract a modest number

38th Conference on Neural Information Processing Systems (NeurIPS 2024).

of interpretable features from high dimensional data, e.g., deriving features of anatomical regions from medical images, because computation time scales super-linearly with the number of features. The preprocessing introduces uncertainty and is often computationally burdensome in itself.

Here we introduce the variational event-based model (vEBM), which enables high dimensional interpretable models through a new computationally efficient approach that avoids the need for dimensionality reduction or manual feature extraction. For example, with image-based models, it enables models that express progression at the pixel-level rather than regional level. To achieve this we reformulate disease progression modelling as the 'transport' of latent disease events to their 'optimal' location in a continuous latent permutation, unlocking benefits from recent advancements in the field of computational optimal transport [42]. Our approach generalises discrete generative models of disease progression, e.g., [1, 20, 5, 36–39, 43], which it obtains as a limit; and it directly infers a continuous probability over events, while the others require costly sampling methods. Crucially, it also facilitates variational inference of the posterior, allowing for substantial gains in computational tractability and hence larger models.

**Related work.**   The closest direct comparisons to the model we propose here are the sequence-based models proposed by [33, 1]. These models underpin both a range of direct applications [20, 5, 23, 24, 12, 21, 25, 27, 29], as well as providing components in higher level models [5, 36, 38, 39]. Like ours, these models require data with only a single time-point per individual. They assume monotonic progression in order to learn a latent sequence of events from such cross-sectional data. However, the models in [33, 1] are severely limited by their computational tractability, and can typically only use a few 100 features at most. In contrast, our new formulation enables this type of model to include several orders of magnitude more features enabling, for example, pixel-level temporal models as we demonstrate here.

Deep-learning based sequence models, using e.g. transformer architectures have recently become popular for models of high dimensional temporal sequences e.g., [41]. However, as with other deep state-space models, e.g., [44, 6] these approaches require vast amounts of data with multiple time points to train, unlike our approach which can be trained on modest datasets (order 100 subjects with observations from a single time-point). Furthermore, the computational power required to train the upstream foundation model, plus the downstream model itself, is order of magnitudes higher than our model, which can run on a single CPU in a matter of minutes.

## 1.1   Contributions

Here we address the problem of how to learn interpretable high dimensional disease progression models efficiently, which is longstanding in the machine learning community.

- We leverage ideas from optimal transport to derive a new generative latent variable model of disease progression, the variational event-based model (vEBM). The vEBM characterises the disease process by a continuous latent permutation of event probabilities, permitting direct inference of event distributions and model uncertainty from mixed feature datasets.

- We define a differentiable variational evidence lower bound (ELBO) and devise a suitable inference scheme to learn the vEBM efficiently from high dimensional data.

- We use synthetic data to demonstrate that the vEBM achieves a factor of $1000\times$ faster inference than baselines, provides better inference accuracy, and is robust to noise.

- We use the vEBM with data from Alzheimer's disease (AD) and age-related macular degeneration (AMD) to obtain, for the first time, pixel-level disease progression events in the brain and eye, and mixed-feature models combining imaging and clinical test score data.

## 2   Variational event-based model

To derive the variational event-based model (vEBM), we first derive a generative latent variable model of disease progression in terms of a latent permutation matrix of events (Section 2.1). Our key methodological contribution is reformulating the generative model in the context of optimal transport; we introduce the relevant mathematical tools to do this in Section 2.2, which we use to derive the limit relationship between the classical EBM and the vEBM in Appendix Section A.3. We then define

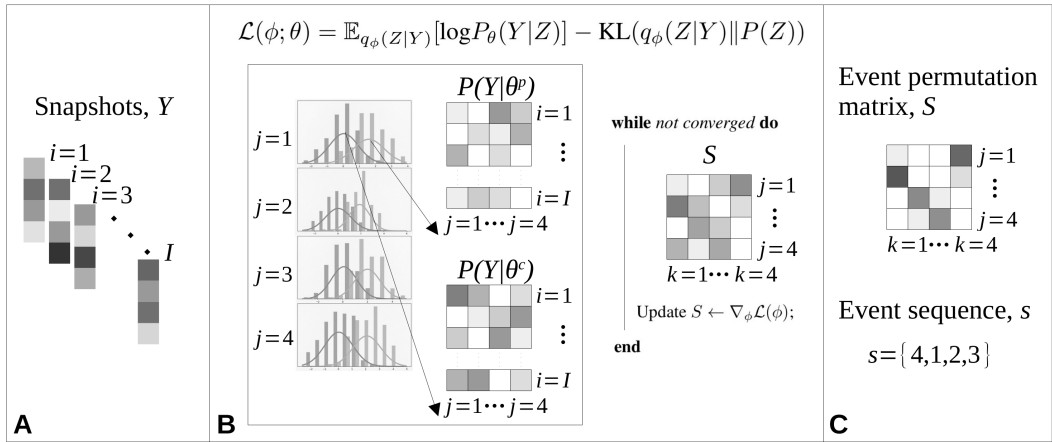

Figure 1: Schematic of the variational event-based model for a toy 4-feature dataset. **A**. The dataset contains snapshots from $I$ individuals, with $j, k = \{1, 2, 3, 4\}$ features and latent events; the features can be of any type and can be incomplete. **B**. Before inference, probabilistic models of normality and abnormality are fit to the dataset, giving the likelihood look-up tables $P(Y|\theta^{p,c})$ (Section 2.1); these are fixed throughout inference, as denoted by the inner box outside the training loop. To infer the permutation matrix $S$ (Section 2.2), the ELBO (Section 2.3) is optimised and $S$ is updated each iteration using the Sinkhorn-Knopp algorithm (Section 2.4). **C**. The resulting hard permutation, $s$, i.e., the disease event sequence, is obtained from $S$, using the Hungarian algorithm. We note that $S$ represents the full distribution of event probabilities, which can be sampled to obtain uncertainty.

our model in a variational inference setting (Section 2.3), devise a suitable inference scheme (Section 2.4), and provide a method for probabilistic individual-level staging using the trained model (Section 2.5). Figure 1 1 provides schematic overview of the vEBM.

## 2.1 Model of disease progression

Consider a generative latent variable model with observed data $Y$ and latent variables $Z = \{S, k, \theta\}$, where $S \in \mathbb{R}_{+}^{N \times N}$ is a latent permutation matrix of $N$ events; $k \in \mathbb{Z}_{+}^{N}$ is the latent state of an individual; and $\theta = \{\theta(1), \theta(2), \theta(3), ..., \theta(J)\}$ are additional model parameters. We write the joint probability as a hierarchical Bayesian model using the chain rule (see Appendix Figure 9 for the graphical model),

$$P(Z, Y) = P(Z) \cdot P(Y|Z) = P(S) \cdot P(\theta) \cdot P(k|S) \cdot P(Y|S, k, \theta). \tag{1}$$

Each element in the permutation matrix, $S$, defines an 'event', which corresponds to a feature becoming measurably abnormal with respect to a reference distribution. Following [1, 20], we parameterise the data likelihood by probability distributions of 'abnormality' (typically from patients, $p$), and 'normality' (typically from controls, $c$) for each feature, and choose to model the distributions for feature $j$ using univariate Gaussian mixture models with mean, $\mu_j$, standard deviation, $\sigma_j$, and mixture weights, $w_j$, so that

$$P(Y_j|\theta_j^{p,c}) \sim \mathcal{N}(\mu_j^{p,c}, \sigma_j^{p,c}, w_j^{p,c}). \tag{2}$$

However any probabilistic characterisation that defines a reference group to anchor the progression is permissible. To enable $S$ to be inferred using data from different individuals at a single time-point ('snapshots'), we make two key assumptions: $i)$ monotonic progression of events at the group level; and $ii)$ a consistent event permutation across the whole population. We note that assumption $ii)$ could be relaxed to permit multiple event permutations (i.e., clusters) within the same population. Then for individuals $i = \{1, 2, 3, ..., I\}$ with observed features $j = \{1, 2, 3, .., J\}$, the model likelihood can be written as (see Appendix A.2 for a full derivation),

$$P(Y|S; \theta) = \prod_{i=1}^{I} \left[ \sum_{k_i=0}^{N} P(k_i|S) \prod_{j=1}^{k_i} P(Y_{i,j}|S, k_i, \theta_{s(j)}^p) \prod_{j=k_i+1}^{J} P(Y_{i,j}|S, k_i, \theta_{s(j)}^c) \right]. \tag{3}$$

Here $s \in \text{Perm}(N)$ is a discrete permutation of $N$ events, corresponding to the hard permutation obtained from $S$ (see the next section); and $\theta_{s(j)} = \theta^p_{s(j)} \cup \theta^c_{s(j)}$ are the patient, $p$, and control, $c$, distribution parameters generating the data for feature $j$ at position $s(j)$ in the permutation. Note that if data are missing, the two likelihoods on the RHS of Equation 3 can be set equal and factorised, i.e., the data can be treated as missing at random. In order to impose no prior information on the permutation ordering, we chose the prior over latent stages to be uniform, $P(k_i|S) \sim \text{Unif}(0, k)$.

## 2.2 Optimal transport for permutations

Under the definition of disease progression given by Equation 3 and using Bayes rule, our posterior is a probability distribution over the sequence of events. Given that permutations are factorial in $N$, the challenge is to make inference of $S$ computationally tractable when the number of features - and hence the number of events - is large (of order $N > 100$). To address this, we propose to frame the learning problem in terms of the 'transport' of disease events to their 'optimal' location in the disease event sequence, $s$, defined by the permutation matrix $S$. Our key methodological contribution is the translation of the model likelihood (Equation 3) to the context of optimal transport; here we provide the necessary background theory to enable us to derive the relationship between $s$ and $S$ in our model.

Optimal transport aims to identify the mass-conserving coupling between two distributions ('transport plan') that minimises the cost required to move (or transform) one into the other [45]. The minimum cost defines a distance between distributions (the Wasserstein distance) and induces a rich underlying geometry on the space of distributions, providing benefits over classical learning techniques such as maximum likelihood. While optimal transport requires solving a computationally expensive linear problem, recent advances have resolved this by substituting the original problem with an entropy regularised version [46], paving the way for its use in learning generative models.

When the couplings are restricted to be permutation matrices, we can leverage the machinery of entropy regularised optimal transport to provide computationally tractable solutions to inferring latent permutations [47]. Here we are interested in learning a latent permutation matrix, $S$, with a corresponding discrete permutation, $s$, such that,

$$\forall (i,j) \in \mathbb{Z}^2_+, \ S_{i,j} = \begin{cases} 1, & \text{if } j = s_i \\ 0, & \text{otherwise.} \end{cases} \tag{4}$$

In the context of permutation matrices as couplings, $S$ belongs to the Birkhoff polytope,

$$\mathcal{B}_N = \{S \in \mathbb{R}^{n \times n}_+ : S_{i,j} \geq 0, \ \sum_j^N S_{i,j} = 1, \ \sum_i^N S_{i,j} = 1\}. \tag{5}$$

The Birkhoff-von Neumann theorem states that $\mathcal{B}_N$ is the convex hull of the set of doubly stochastic (soft) permutation matrices, and that its vertices are the (hard) permutation matrices [48]. The row-column normalisation equality constraints in Equation 5 demand efficient algorithms to solve for $S$. Following [46], we use the Sinkhorn-Knopp algorithm with an entropy regularisation term, $H(S) = -\sum_{i,j} S_{i,j} \log(S_{i,j})$, as an approximation to solving the optimal transport problem,

$$K(X/\tau) = \underset{S \in \mathcal{B}_N}{\text{argmax}} \langle S, X \rangle_F + \tau H(S). \tag{6}$$

Here $K(\cdot)$ is the Sinkhorn-Knopp operator, which maps the positive orthant on to $B_N$ by iteratively normalising rows and columns [49, 50]; $X$ is the unnormalised assignment probability (transportation cost) matrix; and $\tau$ is a temperature parameter, analogous to the temperature-dependent softmax function for discrete categories [51]. In our context, $X$ corresponds to the event likelihood distributions given by Equation 2; as such, we are looking to find the permutation matrix, $S$, that transports event probabilities to their optimal location in the event sequence, $s$. Alternatively, we can think of the relationship as $S$ being the transport plan that permutes event likelihoods in $X$ to their optimal position in the latent event sequence.

To obtain a hard permutation from $S$, we use a result from [47], who showed that $M(X)$, the hard permutation matrix of discrete matches (i.e., the matrix of basis vectors corresponding to the vertices of the Birkhoff polytope), can be obtained as the limit $\tau \to 0$ of the Sinkhorn-Knopp operator,

$$M(X) = \begin{bmatrix} e_{s(0)} \\ \vdots \\ e_{s(N)} \end{bmatrix} = \lim_{\tau \to 0} K(X/\tau). \tag{7}$$

Here $e_n$ are basis (one-hot) vectors of size $N$ with a value of 1 in the $n$-th position and 0 everywhere else. In practice we compute the hard permutation matrix $M(X)$ from $S$ using the Hungarian algorithm [52, 53], which solves the minimum bipartite matching problem in cubic time. We use the relation in Equation 7 to show that the original EBM can be obtained as the temperature limit of the vEBM (see Appendix A.3). To facilitate inference, the value of $\tau$ must be chosen to balance between the limit of a hard permutation, where the gradients are discontinuous (and hence non-differentiable), and a uniform soft permutation, where the gradients are flat (and hence non-informative). A parametric analysis of $\tau$, $\tau_{\mathrm{prior}}$, and the number of Sinkhorn-Knopp iterations, $n_s$, is presented in Appendix A.8.

## 2.3 Variational permutation inference

We approximate the posterior probability, obtained from applying Bayes rule to Equation 3, using variational inference [54], and define the evidence lower bound (ELBO). To enable differentiability of the ELBO, we parameterise our variational prior and posteriors using the Gumbel-Sinkhorn distribution, $G(X, \tau)$, with a matrix, $\epsilon$, of i.i.d. Gumbel noise,

$$G(X, \tau) \sim K((X + \epsilon)/\tau). \tag{8}$$

The Gumbel-Sinkhorn distribution effectively implements the reparameterisation trick [55] for permutations, and in the limit $\tau \to 0$ it has been shown to converge to the Gumbel-Matching distribution, the equivalent of the Gumbel-Sinkhorn distribution for hard matchings [47]. We choose a uniform prior over permutations, $G(X = 0, \tau_{\mathrm{prior}})$, and for the posterior, $G(X, \tau; \phi)$, with parameters $\phi$. We seek to optimise the corresponding ELBO,

$$
\begin{aligned}
\log P(Y) \geq \mathcal{L}(\phi; \theta) &= \mathbb{E}_{q_\phi(Z|Y)}[\log P_\theta(Y|Z)] - \mathrm{KL}(q_\phi(Z|Y)\|P(Z)) \\
&= \mathbb{E}_{q_\phi(Z|Y)}[\log P_\theta(Y|Z)] - \mathrm{KL}(G_\phi(X, \tau)\|G(X = 0, \tau_{\mathrm{prior}})).
\end{aligned} \tag{9}
$$

The Kullback-Leibler (KL) divergence term on the RHS of Equation 9 is intractable, but can be rewritten as $\mathrm{KL}((X + \epsilon)/\tau\|\epsilon/\tau_{\mathrm{prior}})$ by substituting $(X + \epsilon)/\tau$ for $Z$ and estimated using random sampling [47]. For completeness, we restate the full term derived by [47] in Appendix A.4.

## 2.4 Inference scheme

To optimise the ELBO we use Adam [56] with $n_{\mathrm{opt}} = 200$ iterations and a learning rate of 0.1. Temperature hyperparameters $\tau$ and $\tau_{\mathrm{prior}}$ were set to 1 for all experiments, except the mixed events (Section 3.3.3), where $\tau = 1E3$. We found setting $\epsilon = 0$ during inference gave the fastest and most accurate estimate of $S$, at the expense of not allowing for direct propagation of uncertainty. While uncertainty estimation is not the focus of this paper, we do provide some examples of setting $\epsilon$ non-zero in Appendix A.7. Pseudo-code for the full inference scheme is given in Appendix Algorithm 1.

## 2.5 Probabilistic staging

We can use the trained model to obtain an individual-level likelihood distribution over stages, i.e., the likelihood at each state $k$ given by Equation 3, where stage $k = n$ corresponds to the first $n$ events occurred and the remaining $N - n$ events not occurred. Here we simply take the maximum likelihood stage for each individual, but alternative summary statistics could be used.

# 3 Experiments

## 3.1 Baselines

We consider two baselines; i) the original EBM [1]; and the Alzheimer's Disease Probabilistic Cascades (ALPACA) model [33], both of which use maximum likelihood to estimate the ordered sequence, $s$. The EBM learns $s$ using gradient descent and Markov Chain Monte Carlo (MCMC) sampling. The ALPACA model instead defines $s$ as the central permutation of a Mallows model [57], with a density over permutations given by $p(s) \sim \exp(-\lambda d(s, s_0))$, where $\lambda$ scales the spread around the central ordering, $s_0$, and $d(s, s_0) = \Sigma_{i=1}^{N}|s(i) - s_0(i)|$ is the distance between permutations. The ALPACA model learns $s$ using expectation-maximisation (EM) and Gibbs sampling. We use the default parameters; for the EBM, $10^3$ gradient descent iterations with 10 initial seeds, and $10^6$ MCMC samples; for ALPACA, 10 EM iterations and 100 Gibbs samples.

## 3.2 Synthetic data

To enable comparison between our model and the baselines, we simulate data generated by an ordered sequence, $s$, according to the limit version of Equation 3 (see Appendix A.3). In brief, $s$ is randomly initialised and individuals are assigned a stage with uniform probability, reflecting that individuals can be observed at any disease stage. Individuals are assigned as either controls or patients according to an arbitrary threshold on the disease stage (here we choose the lowest 20% stages as controls). Feature data for each individual are then generated from the Gaussian models of normal and abnormal feature values, depending on their stage, with zero means for the control distributions, random uniform means for the case distributions, and variable standard deviations for both controls and cases set to achieve a desired level of noise. Exact parameter values and code to generate synthetic data is given in the GitHub repository. Here we repeat performance evaluation over 10 simulated datasets for each experiment to support statistical significance tests.

### 3.2.1 Faster inference

Figure 2 shows the runtime for the vEBM and baselines for three experiments ($I = 100$, $J = 10$; $I = 1000$, $J = 100$; $I = 2000$, $J = 200$). We do not consider $J > 200$ here, as the baselines become intractable, but Figure 2 clearly illustrates the unique computational ability of the vEBM to work with much larger $J$, as we demonstrate throughout this section. The vEBM is a factor of 1000 times faster for the $J = 200$ experiment; this factor would only increase for larger models.

### 3.2.2 Improved accuracy and robustness to noise

Figure 3 shows the effect of increasing aleatoric (measurement) noise levels on inference accuracy, as measured by the Kendall's tau [58] between the true and inferred sequences. The vEBM outperforms or is comparable to the baselines in all datasets and noise settings, except for $I = 100, J = 1000$ and $\sigma = 0.1$; this is expected, because at low noise and smaller numbers of features the EBM's MCMC sampling should find the global minimum, while the vEBM will always have some uncertainty due to its variational approximation. Statistical significance was obtained at $p < 0.001$ using unpaired t-tests (note that only one datapoint is shown for ALPACA at $J = 100$, and none at $J = 200$, due to computational intractability). We highlight that the metric is sensitive to any departure from the correct ordering, even by a single sequence position; accordingly the visual correlation between the true and inferred

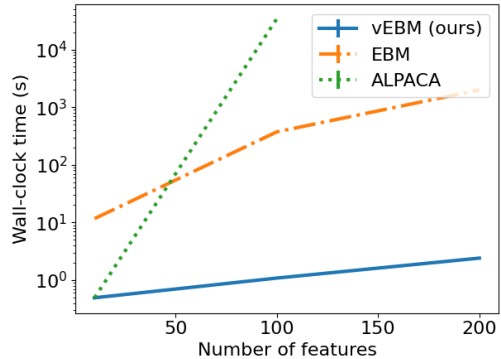

Figure 2: Speed of inference as a function of model size, for the vEBM and baselines. Note that there is no datapoint for ALPACA at $J = 200$ due to computational intractability.

sequence remains high, e.g., even at the highest noise ($\sigma = 1$, corresponding to a 1:1 signal:noise ratio), as the relationship is still approximately diagonal. Additional examples in other datasets and when setting the Gumbel noise, $\epsilon$, non-zero are given in Appendix A.6, A.7.

## 3.3 Alzheimer's disease data

We use pre-processed tensor-based morphometry (TBM) data from the Alzheimer's Disease Neuroimaging Initiative (ADNI) study, a longitudinal observational study of AD. TBM data are derived from structural magnetic resonance imaging (MRI) data and represent voxel-level maps of intensity gradients with respect to a reference healthy brain template, providing a standardised measure of voxel-level volume loss (or gain) between individuals. The TBM dataset we use here is comprised of cross-sectional TBM maps from 816 individuals (299 controls, 399 mild cognitive impairment, 188 AD) [59]. In Section 3.3.3 we also use three cognitive test scores – Mini-Mental State Examination (MMSE); Clinical Rating Dementia scale Sum of Boxes (CDRSB); Rey Auditory Verbal Learning Test (RAVLT). Both datasets are available to download for users with an ADNI account

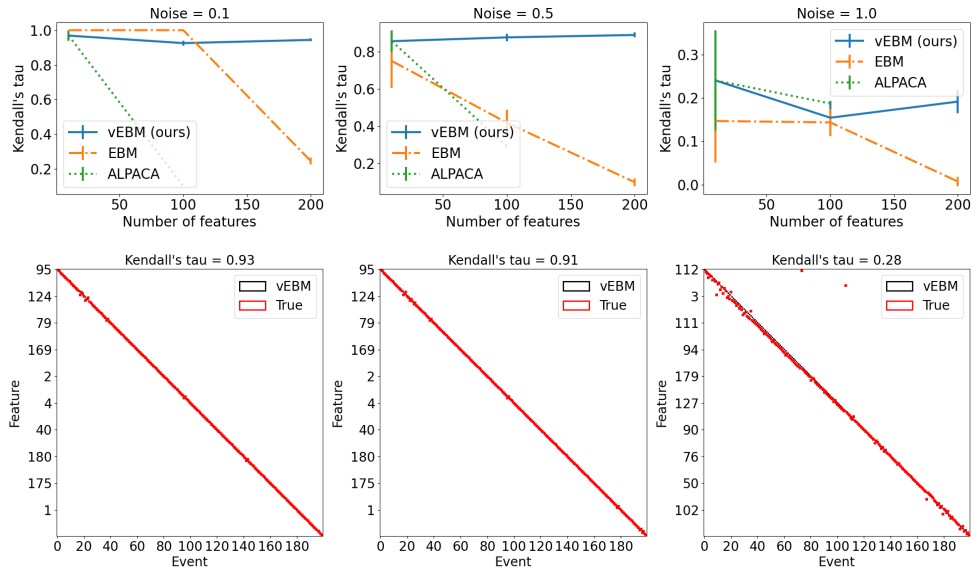

Figure 3: Accuracy of inference as a function of aleatoric noise, obtained by the vEBM from synthetic data with low, medium, and high noise levels (left: $\sigma = 0.1$; middle: $\sigma = 0.5$; right: $\sigma = 1$). Top row: Kendall's tau distance between the inferred and true sequences as a function of model size (number of features). Standard errors of the mean are shown, from 10 repeats per experiment. Bottom row: example positional variance diagrams. The vertical axis lists the sequence of events inferred by the vEBM with the earliest event (order position 1) at the top. The true sequence is overlaid as red squares. Datasets have $I = 2000$ individuals and $J = 200$ features.

(`https://adni.loni.usc.edu/data-samples/access-data/`, data collections: "TBM Jacobian Maps MDT-SC"; "Tadpole Challenge").

### 3.3.1 Pixel-level disease progression events in AD

We apply the vEBM to TBM data from ADNI to reveal the first pixel-level sequence of disease events in AD (Figure 4). We do not include the baselines here due to computational intractability (as demonstrated in Section 3.2). The pattern of change represented by the vEBM sequence recapitulates known large-scale changes due to AD; initial change in the ventricles, followed by other sub-cortical changes, then changes across the cortex [60]. Moreover, the vEBM finds a detailed pattern of grey and white matter changes throughout the sequence, providing new small-scale insights into AD aetiology that have not previously been possible, which we explore further in the next section.

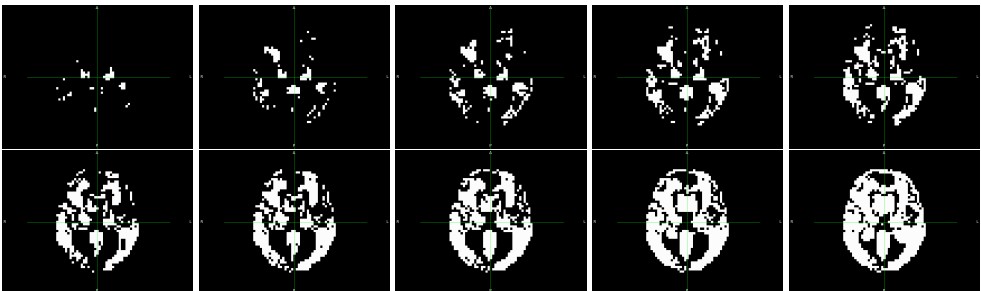

Figure 4: Pixel-level disease progression sequence in AD obtained by the vEBM. White pixels correspond to events that have occurred by the corresponding point of the sequence. The figure shows 10 sequence positions at uniform steps of 100 across the total of 1344, with the top left figure corresponding to position 50 (the first 50 events have occurred) and the bottom right to position 950. Images were made from the vEBM output using 3D Slicer (`https://www.slicer.org/`).

### 3.3.2 Segmentation-based interpretation of pixel-level events

To evaluate our ADNI pixel-level model with respect to previous analyses that have used segmented regional brain areas, we map the vEBM pixel-level events post hoc to pixel-level labels obtained from the FreeSurfer segmentation of the reference template (Figure 5). Note that the regions shown are a subset of the total regions available from the FreeSurfer segmentation tool, which were chosen according to those that were sufficiently represented in terms of number of pixels in the 2D slice that was used to train the model ($N > 10$). Also note that the number of points on each trajectory / line corresponds to the number of pixels available in each region; e.g., there are few pixels in "Putamen" ($N = 11$), an intermediary number in "Thalamus-Proper" ($N = 43$), and many in "Cerebral-Cortex" ($N = 299$), corresponding to their relative sizes (areas) in the 2D slice.

Our findings are in broad agreement with previous results; sub-cortical changes (Thalamus-Proper, Putamen, Hippocampus) are earliest, followed by cortical (Cerebral-Cortex) and white matter (Cerebral-White-Matter), and finally ventricular change (Lateral-Ventricle, VentralDC). However, our model provides much more fine-grained insights than, e.g., [20], we now obtain continuous trajectories of change, which capture interesting non-linearities, e.g., in the Thalamus-Proper, Brain-Stem, and Lateral-Ventricle; this contrasts with the more linear changes in the Hippocampus, Cerebral-Cortex, and Cerebral-White-Matter.

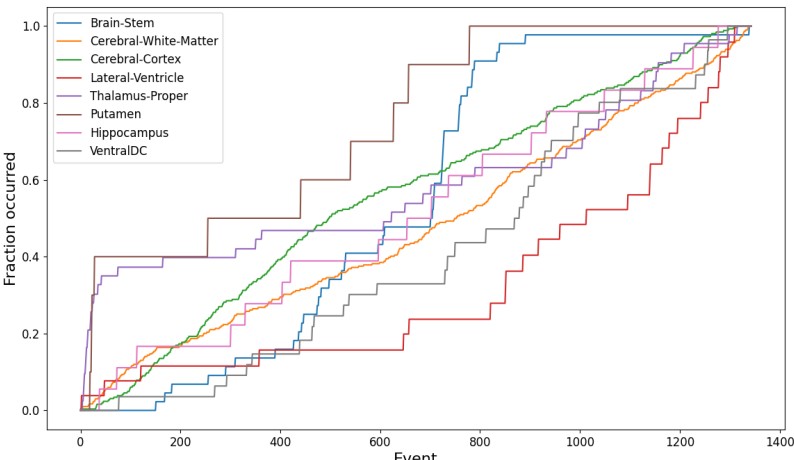

Figure 5: Trajectories of regional brain areas in our ADNI cohort, obtained by mapping the vEBM pixel-level events to pixel-level labels obtained from the FreeSurfer segmentation of the reference template. The horizontal axis shows the event number (from 0 – 1344), and the vertical axis shows the fraction of pixel-events that have occurred in each regional brain area at the corresponding event number, as defined by the vEBM event sequence.

### 3.3.3 Mixed feature disease progression events in AD

The vEBM is not limited to modelling only image-based features, which we demonstrate by including three cognitive test score features (MMSE, CDRSB, RAVLT) and re-training the model. Figure 6 represents the spatio-(pseudo)-temporal pixel event topology obtained by the vEBM as a 2D histogram, and shows the position of the cognitive events by vertical lines. We calculate the spatial distribution of pixel events according to their Euclidean distance from the centre of the image. The colour denotes the number of pixel events in each histogram bin, e.g., in the first bin of events (the first column), we can see the density of pixel events occurring as a function of the distance from the centre. The pixel event topology shows the earliest events near the centre of the brain, as expected [60], before spreading out across the brain; these events are interleaved with cognitive events, which occur across the latter two thirds of the progression. This interleaving suggests that the vEBM could be used to provide fine-grained staging in between cognitive events, e.g., for stratification in clinical trials. Interestingly, the pixel event topology is asymmetric about the central axis of the brain in the

early stages, suggesting that the vEBM can identify subgroups of individuals who display asymmetric progression, which has previously been reported in small groups of people with AD, e.g., [61].

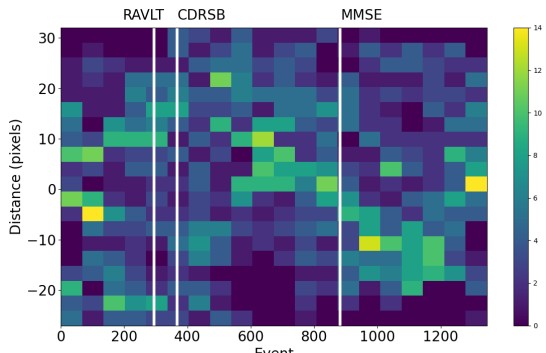

Figure 6: Mixed pixel and cognitive feature event topology in AD found by the vEBM. The vertical axis shows the distance from the centre of the image to the pixel event, and the horizontal axis shows the event ordering obtained by the vEBM. Note that the cognitive events, denoted by vertical lines, are assigned an arbitrary distance of zero.

## 3.4 Age-related macular degeneration data

We use pre-processed optical coherence tomography (OCT) data from the Duke University (DU) Ophthalmology 2013 dataset, a cross-sectional study of AMD [62]. The OCT data represent thickness maps for retinal pigment epithelium and drusen complex (RPEDC), a marker of AMD progression. The OCT dataset is comprised of 384 individuals (115 controls, 269 AMD), and is publicly available to download: `https://duke.app.box.com/s/l80j6ziooeyy1eeo7edy0il32zbyyzbg`. To select only pixels with disease signal, we remove pixels with an effect size less than 4 on a pixel-wise t-test between the control and AMD groups.

### 3.4.1 Pixel-level disease progression events in AMD

We apply the vEBM to OCT data from the DU cohort to reveal the first pixel-level sequence of disease events in AMD (Figure 7). The density of RPEDC spread around the centre of the eye reflects previous observations [62], and the vEBM provides a much finer-detailed progression pattern.

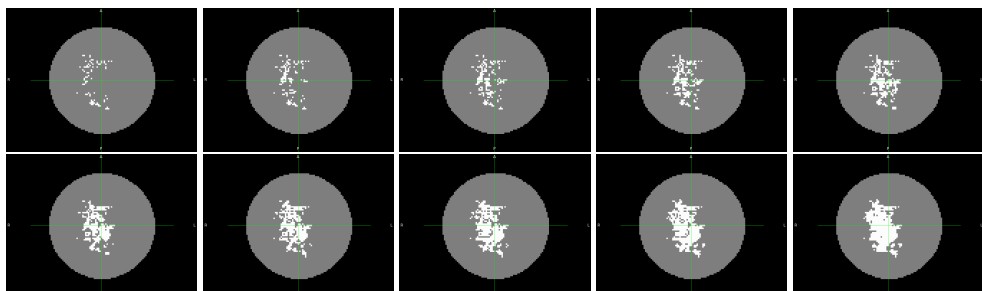

Figure 7: Pixel-level disease progression sequence in AMD obtained by the vEBM. White pixels correspond to events that have occurred by the corresponding point of the sequence. We have selected 10 sequence positions at uniform steps of 50 across the total of 537 in the full sequence, with the top left figure corresponding to position 80 and the bottom right to position 530. Images were made from the vEBM output using 3D Slicer (`https://www.slicer.org/`).

## 3.5 Prediction of AD and AMD stage

Figure 8 shows the stage distribution for individuals in the ADNI and DU cohorts using the vEBM trained on mixed and pixel-only data, respectively. We find a fine-grained distribution of individual-level stages that reflects the clinical labels, demonstrating the utility of the vEBM for stratification tasks, e.g., to select cohorts for clinical trials [63].

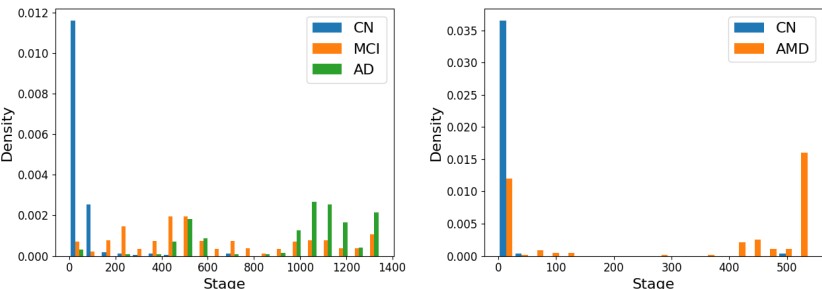

Figure 8: Individual stages obtained by the vEBM in AD (left) and AMD (right). CN: control; MCI: mild cognitive impairment; AD: diagnosed AD; AMD: diagnosed AMD.

## 4 Discussion

We introduced the vEBM, a novel optimal transport formulation for discrete disease progression models that scales to hitherto impossible numbers of events. It allows pixel-level visualisation of the order of pathology appearance in chronic disease, as demonstrated for the first time in AD and AMD.

### 4.1 Limitations

Here we did not fully explore model uncertainty, which as a Bayesian model, the vEBM can estimate directly; we reserve this for future work. We acknowledge that a model ordering several thousand events is not fully identifiable with only a few hundred snapshots, but the orderings we obtain are still highly meaningful, as demonstrated by our results in AD and AMD. Moreover, our optimal transport formulation naturally lends itself to feature-sparsification [64], allowing redundant events to be grouped together. While the vEBM can in principal be directly applied to raw image data, pre-processing of images to a common reference frame (i.e., image registration) is necessary to facilitate comparison between individuals; however pre-processing is necessary for any data type to be in a common reference frame (or scale). We do not explicitly account for feature-wise covariance in the model, e.g., in image-based data we would expect a high degree of collinearity between neighbouring pixels; this could be addressed by including an additional term that imposes local structure, e.g., a Markov random field [35]. Finally, we note that the main limitation on the current formulation is not computational tractability but computer memory due to dense matrix operations, which could be alleviated using, e.g., sparse matrix representation.

### 4.2 Broader impacts

The vEBM enables disease progression modelling at scale in multiple areas of medical imaging, not only the modalities demonstrated in this paper; such as other MRI modalities e.g., diffusion weighted imaging, microstructure modelling, connectivity; other imaging modalities, e.g., positron emission tomography, computed tomography, X-rays, ultrasound; and non-radiological imaging modalities, e.g., microscopy. Furthermore, the vEBM can run quickly on a relatively low-spec computer without the need for GPU infrastructure, making it accessible to research labs – and potentially clinics – that have limited resources, while further minimising its carbon impact by reducing compute time. In addition, it provides a new, more powerful, model for each component of mixture subtype models, e.g., [5], which currently uses a variant of the basic EBM. Such models are highly influential in stratifying patients into disease subgroups for more precise clinical trials and treatment deployment.

## Acknowledgments and Disclosure of Funding

PAW would like to acknowledge useful discussions with the Sussex PAL Book Club and logistical support from Siobhán and Lorcán. DCA would like to acknowledge support from Wellcome Trust grant 221915 and the NIHR UCLH Biomedical Research Centre. Both authors acknowledge support from the Wellcome Leap 1kD project. Neither author have competing financial interests to declare.

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

# A Appendix / supplemental material

## A.1 Compute information

The analyses presented here were performed either on a laptop PC with a single AMD Ryzen 7 PRO 6860Z CPU with 32GB RAM, or a desktop PC with a single AMD Ryzen Threadripper PRO 5975WX CPU with 270GB RAM (we note that we also provide a GPU implementation of the code, but we do not use it here to allow fair comparison with the baselines). Model training for the synthetic data analysis took 48 hours (wall-clock time) in total for all experiments. Model training for the AD analysis took 5 minutes. Model training for the AMD analysis took 1 minute. No pre-training of the model was performed. Approximately 48 hours CPU compute, and less than one hour on a GPU, was performed for code testing and preliminary experiments that are not included in this paper. Python code to reproduce all results presented here is available from the first author's GitHub repository[1].

## A.2 Derivation of vEBM

Starting from the joint probability (Equation 1), we use the assumption of independence between measured features $j = \{1, 2, 3, ..., J\}$ to write the model likelihood,

$$P(Y_i|k_i, S, \theta) = \prod_{j=1}^{J} P(Y_{i,j}|k_i, \theta_{s(j)}, S). \tag{10}$$

Using the chain rule, the joint probability distribution over $Y$ and $k$ can be factorised as,

$$P(Y_i, k_i|S, \theta) = P(k_i|S) \prod_{j=1}^{J} P(Y_{i,j}|k_i, \theta_{s(j)}, S). \tag{11}$$

For the likelihood model (Equation 10), here we choose a two-component Gaussian mixture model (though as noted in the main text, any probabilistic model can be chosen),

$$\prod_{j=1}^{J} P(Y_{i,j}|k_i, \theta_{s(j)}, S) = \prod_{j=1}^{k_i} P(Y_{i,j}|k_i, \theta_{s(j)}^p, S) \prod_{j=k_i+1}^{J} P(Y_{i,j}|k_i, \theta_{s(j)}^c, S). \tag{12}$$

Substituting Equation 12 into Equation 11 and marginalising over $k_i$, we have,

$$P(Y_i|S, \theta) = \sum_{k_i=0}^{N} P(k_i|S, \pi) \prod_{j=1}^{k_i} P(Y_{i,j}|k_i, \theta_{s(j)}^p, S) \prod_{j=k_i+1}^{J} P(Y_{i,j}|k_i, \theta_{s(j)}^c, S). \tag{13}$$

Here we chose the prior over latent stages, parametrised by hidden variable $\pi$, to be uniform and constant, $P(k_i|S; \pi) \sim \text{Unif}(0, k)$. Finally, assuming independence between measurements from different individuals $i$, we can write the following expression for the total likelihood,

$$P(Y|S, \theta) = \prod_{i=1}^{I} \left[ \sum_{k_i=0}^{N} P(k_i|S, \pi) \prod_{j=1}^{k_i} P(Y_{i,j}|k_i, \theta_{s(j)}^p, S) \prod_{j=k_i+1}^{J} P(Y_{i,j}|k_i, \theta_{s(j)}^c, S) \right]. \tag{14}$$

Bayes' theorem can now be used to obtain the posterior over $S$.

## A.3 Posterior limit

We use the limit relation in Equation 7 to reparametrise the model likelihood (Equation 3) in terms of a discrete permutation, $s$,

$$P(Y|s, \theta) = \lim_{\tau \to 0} P(Y|S, \theta; \tau). \tag{15}$$

---

[1] https://github.com/pawij/vebm

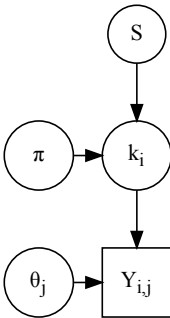

Figure 9: Graphical model of the variational event-based model (vEBM). Hidden variables are denoted by circles and observations by squares. $S$: event permutation matrix; $\pi$: initial probability vector (fixed to uniform distribution); $k_i$: disease state for individual $i$; $\theta_j$: distribution parameters for biomarker $j$; $Y_{i,j}$: observed data.

To see this, we take the first likelihood term in the RHS of Equation 3 and write it in matrix form,

$$
\begin{aligned}
P(Y_{i,j}|s, \theta^p_{s(j)}) &= \begin{bmatrix} p(Y_{0,0}|\theta^p_{s(0)}) & \cdots & p(Y_{0,J}|\theta^p_{s(J)}) \\ \vdots & \ddots & \vdots \\ p(Y_{I,0}|\theta^p_{s(0)}) & \cdots & p(Y_{I,J}|\theta^p_{s(J)}) \end{bmatrix} \\
&= \begin{bmatrix} p(Y_{0,0}|\theta^p_0) & \cdots & p(Y_{0,J}|\theta^p_J) \\ \vdots & \ddots & \vdots \\ p(Y_{I,0}|\theta^p_0) & \cdots & p(Y_{I,J}|\theta^p_J) \end{bmatrix} \cdot \begin{bmatrix} e_{s(0)} \\ \vdots \\ e_{s(J)} \end{bmatrix}^T \\
&= P(Y_{i,j}|\theta^p_j) \cdot M(X)^T \\
&= P(Y_{i,j}|\theta^p_j) \cdot \lim_{\tau \to 0} K(X/\tau)^T,
\end{aligned}
\tag{16}
$$

where we have used the limit relation from Equation 7 between lines three and four. The same steps can be applied to the second likelihood term on the RHS of Equation 3, $P(Y_{i,j}|S, \theta^c_{s(j)})$, to obtain the full likelihood reparametrisation in $s$.

## A.4 Kullback-Leibler divergence

For completeness, we restate the KL divergence term derived by [47],

$$
\mathrm{KL}(X + \epsilon)/\tau \| \epsilon/\tau_{\text{prior}}) = N^2(\log(\tau/\tau_{\text{prior}}) - 1 + \gamma(\tau_{\text{prior}}/\tau - 1)) + S_1 \tau_{\text{prior}}/\tau + S_2 \Gamma(1 + \tau_{\text{prior}}/\tau),
\tag{17}
$$

where $S_1 = \Sigma_{i,j} x_{i,j}$ and $S_2 = \Sigma_{i,j} \exp(-x_{i,j} \tau_{\text{prior}}/\tau)$. Full a full derivation see [47], Supplementary Methods B.3.

## A.5 Inference scheme

Pseudo-code for the vEBM inference scheme is given in Algorithm 1.

## A.6 Positional variance diagrams

Figures 10 and 11 show positional variance diagrams for datasets with $I = 100, J = 10$, and $I = 1000, J = 100$, individuals and features, respectively.

## A.7 Uncertainty estimation

Figures 12, 13, 14 show equivalent positional variance diagrams to Figures 3, 10, 11, but with the Gumbel noise term, $\epsilon$, set to non-zero. The corresponding uncertainty obtained from 1000 random samples of the posterior is shown by greyscale shading on the positional variance diagrams.

**Algorithm 1:** Pseudo-code of the variational event-based model (vEBM) inference scheme. For the experiments in this paper we use $n_s = 20$ Sinkhorn-Knopp iterations, unless otherwise stated.

---

**Input** : $Y, \tau, \tau_{\text{prior}}, n_{\text{opt}}, n_s, \epsilon$, learning rate
**Output**: $\theta, s$
`// fit mixture models`
Compute $P(Y|\theta^{p,c}) \leftarrow \text{EM}(Y, \theta^{p,c})$;
`// infer permutation matrix`
Initialise $S$;
**for** $n_{opt}$ **do**
    Sample $\epsilon$; `// matrix of Gumbel noise`
    **for** $n_s$ *iterations* **do**
        Update $S \leftarrow K(X + \epsilon/\tau)$; `// Sinkhorn-Knopp algorithm`
    **end**
    Compute $\mathbb{E}_{q_\phi(Z|Y)}[\log P_\theta(Y|Z)] - \text{KL}(q_\phi(Z|Y)\|P(Z))$; `// ELBO`
    Update $S \leftarrow \nabla_\phi \mathcal{L}(\phi)$;
**end**
`// compute sequence`
Compute $s \leftarrow S \cdot [0, 1, 2, ..., N]^T$;

---

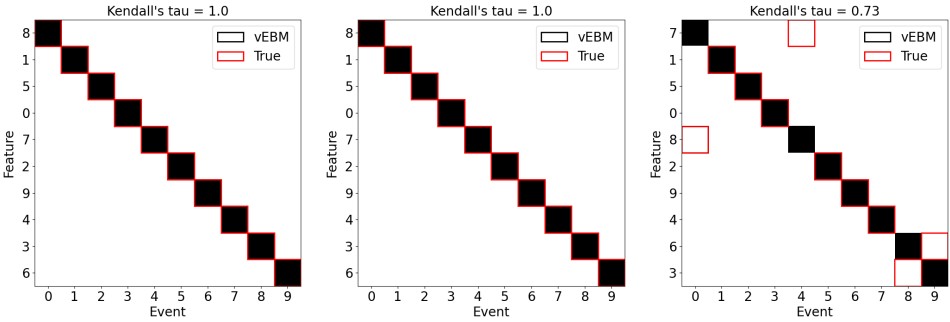

Figure 10: Example positional variance diagrams obtained by the vEBM from synthetic data with low, medium, and high noise levels (left: $\sigma = 0.1$; middle: $\sigma = 0.5$; right: $\sigma = 1$). The vertical axis lists the sequence of events inferred by the vEBM with the earliest event (order position 1) at the top. The true sequence is overlaid as red squares. Datasets have $I = 100$ individuals and $J = 10$ features.

## A.8 Hyperparameter study

Tables 1,2,3 show evaluation metrics for varying combinations of the temperature hyperparameter, $\tau$, and its prior, $\tau_{\text{prior}}$. Table 4 shows the same metrics for constant $\tau, \tau_{\text{prior}}$ and extreme values for the number of Sinkhorn-Knopp iterations, $n_s$. Synthetic data with noise, $\sigma = 0.5$, were used.

Table 1: $\tau$ hyperparameter study, for $\tau_{\text{prior}} = 1$; learning rate = 0.1; $n_s = 10$; $n_{\text{opt}} = 100$.

|  | $\tau = 0.1$ | | | $\tau = 10.0$ | | |
|---|---|---|---|---|---|---|
| $I \times J$ | $100 \times 10$ | $1000 \times 100$ | $2000 \times 200$ | $100 \times 10$ | $1000 \times 100$ | $2000 \times 200$ |
| Kendall's tau | 0.69 | 0.01 | 0.05 | 1.0 | 0.81 | 0.54 |
| Frac. correct | 0.8 | 0.06 | 0.03 | 1.0 | 0.84 | 0.62 |

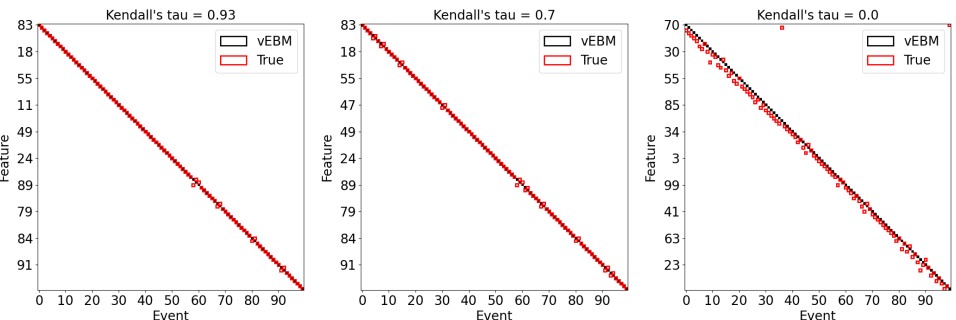

Figure 11: Example positional variance diagrams obtained by the vEBM from synthetic data with low, medium, and high noise levels (left: $\sigma = 0.1$; middle: $\sigma = 0.5$; right: $\sigma = 1$). The vertical axis lists the sequence of events inferred by the vEBM with the earliest event (order position 1) at the top. The true sequence is overlaid as red squares. Datasets have $I = 1000$ individuals and $J = 100$ features.

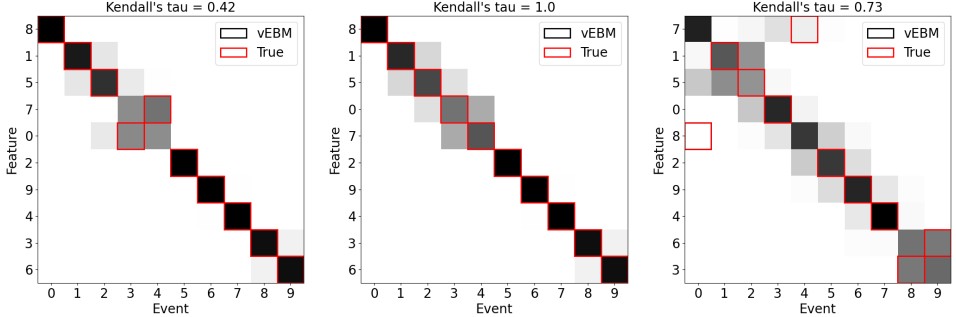

Figure 12: Example positional variance diagrams obtained by the vEBM from synthetic data with low, medium, and high noise levels (left: $\sigma = 0.1$; middle: $\sigma = 0.5$; right: $\sigma = 1$). The vertical axis lists the sequence of events inferred by the vEBM with the earliest event (order position 1) at the top. The matrix shows uncertainty in the ordering: dark squares on the diagonal indicate high certainty of event position; lighter colors and off-diagonal squares indicate uncertainty in the event position. The true sequence is overlaid as red squares. Datasets have $I = 100$ individuals and $J = 10$ features.

Table 2: $\tau$ hyperparameter study, for $\tau_{\text{prior}} = 0.1$; learning rate = 0.1; $n_s = 10$; $n_{\text{opt}} = 100$.

| | $\tau = 0.1$ | | | $\tau = 10.0$ | | |
|---|---|---|---|---|---|---|
| $I \times J$ | $100 \times 10$ | $1000 \times 100$ | $2000 \times 200$ | $100 \times 10$ | $1000 \times 100$ | $2000 \times 200$ |
| Kendall's tau | 1.0 | 0.87 | 0.5 | 1.00 | 0.79 | 0.14 |
| Frac. correct | 1.0 | 0.94 | 0.54 | 1.00 | 0.8 | 0.24 |

Table 3: $\tau$ hyperparameter study, for $\tau_{\text{prior}} = 10.0$; learning rate = 0.1; $n_s = 10$; $n_{\text{opt}} = 100$.

| | $\tau = 0.1$ | | | $\tau = 10.0$ | | |
|---|---|---|---|---|---|---|
| $I \times J$ | $100 \times 10$ | $1000 \times 100$ | $2000 \times 200$ | $100 \times 10$ | $1000 \times 100$ | $2000 \times 200$ |
| Kendall's tau | 0.07 | 0.1 | 0.01 | 0.69 | -0.09 | 0.08 |
| Frac. correct | 0.1 | 0.0 | 0.01 | 0.8 | 0.06 | 0.04 |

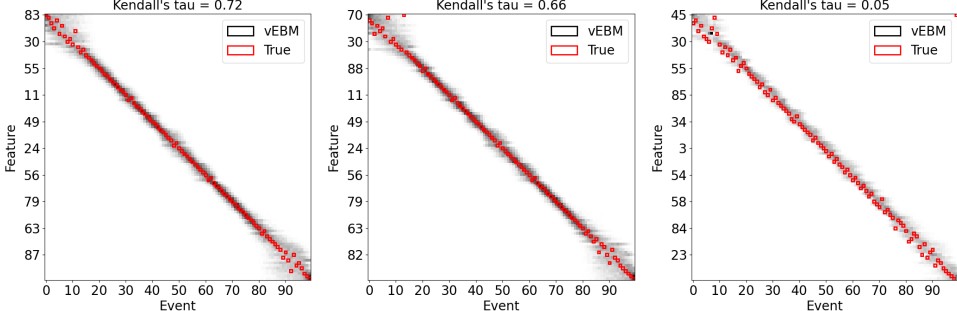

Figure 13: Example positional variance diagrams obtained by the vEBM from synthetic data with low, medium, and high noise levels (left: $\sigma = 0.1$; middle: $\sigma = 0.5$; right: $\sigma = 1$). The vertical axis lists the sequence of events inferred by the vEBM with the earliest event (order position 1) at the top. The matrix shows uncertainty in the ordering: dark squares on the diagonal indicate high certainty of event position; lighter colors and off-diagonal squares indicate uncertainty in the event position. The true sequence is overlaid as red squares. Datasets have $I = 1000$ individuals and $J = 100$ features.

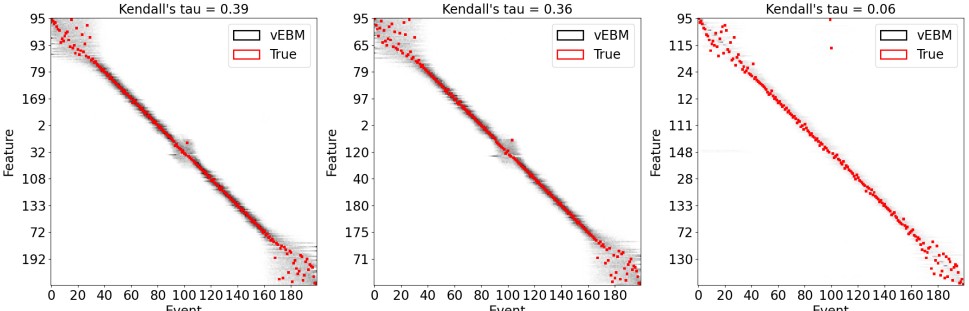

Figure 14: Example positional variance diagrams obtained by the vEBM from synthetic data with low, medium, and high noise levels (left: $\sigma = 0.1$; middle: $\sigma = 0.5$; right: $\sigma = 1$). The vertical axis lists the sequence of events inferred by the vEBM with the earliest event (order position 1) at the top. The matrix shows uncertainty in the ordering: dark squares on the diagonal indicate high certainty of event position; lighter colors and off-diagonal squares indicate uncertainty in the event position. The true sequence is overlaid as red squares. Datasets have $I = 2000$ individuals and $J = 200$ features.

Table 4: $n_s$ hyperparameter study, for $\tau, \tau_{\text{prior}} = 1.0$; learning rate $= 0.1$; $n_s = 10$; $n_{\text{opt}} = 100$.

| $I \times J$ | $n_s = 1$ | | | $n_s = 100$ | | |
|---|---|---|---|---|---|---|
| | $100 \times 10$ | $1000 \times 100$ | $2000 \times 200$ | $100 \times 10$ | $1000 \times 100$ | $2000 \times 200$ |
| Kendall's tau | 0.07 | 0.16 | 0.07 | 1.0 | 0.79 | 0.94 |
| Frac. correct | 0.4 | 0.19 | 0.12 | 1.0 | 0.83 | 0.95 |

