# OpenReview forum: "Unscrambling disease progression at scale: fast inference of event permutations with optimal transport"
_NeurIPS.cc/2024/Conference — NeurIPS 2024 poster_

### Official Review · Reviewer_5r2K · 2024-07-01

**Soundness:** 2
**Presentation:** 2
**Contribution:** 2
**Rating:** 5
**Confidence:** 2

**Summary:**

This paper proposes to use Sinkhorn algorithm to compute optimal transport in order to speed up disease progression that was previously computationally prohibited. This method enables disease progression models with higher dimensionality in features as well as 1000x faster inference speed. Authors provide experiments on Alzheimer’s disease and age-related macular degeneration at pixel-level disease progression events.

**Strengths:**

* Interesting experimental results showing wall-clock improvement on synthetic dataset
* Idea is well motivated and easy to understand

**Weaknesses:**

* On the Pixel-level disease experiments, it's hard to judge how realistic the simulated diseases progressions are without quantitative comparison with baseline/groundtruth, such as co-occurances of events. Since it's a scientific study on method compared with baseline, the claims become unfalsifiable if such setup is provided.
* Overall, this work gives audience the impression of application of Sinkhorn algorithm. Given the lack of my domain expertise in medical science, it's hard for me to judge the novelty of the problem setting. However, method-wise, novelty is limited.
* Practicality of the method in real life problems: Although the claim about "high dimension", which goes up to 200 features, however modern medical MRI machines can easily capture high resolution images with magnitude higher number features/voxels. It's unclear how this method will scale in the more relevant settings.

**Questions:**

In both figure 4, and figure 6, it mentions, " White pixels correspond to events that have occurred; black not yet occurred". However, there doesn't seem to be any black spots?

**Limitations:**

The limitation has been discussed at end of the paper.

---

> ### Author Rebuttal · Authors · 2024-08-06
>
> Weaknesses point 1 (“On the Pixel-level experiments…”)
>
> Yes, it is a good idea to compare with regional-level results, and one we would have included given the time. We have now included an additional analysis in the “Author Rebuttal” PDF, where we used the FreeSurfer segmentation tool to obtain pixel-level anatomical labels, which we mapped to our “pixel” events to enable comparison with regional volumes (Figure 1 in the uploaded PDF).
>
> Indeed what we see is broadly what we’d expect from previous results – sub-cortical changes (Thalamus-Proper, Putamen, Hippocampus) are earliest, followed by cortical (Cerebral-Cortex) and white matter (Cerebral-White-Matter), and finally ventricular change (Lateral-Ventricle, VentralDC). However our model provides much more fine-grained insights – we now obtain continuous trajectories of change, which capture interesting non-linearities, e.g., in the Thalamus-Proper, Brain-Stem, and Lateral-Ventricle; this contrasts with the more linear changes in the Hippocampus, Cerebral-Cortex, and Cerebral-White-Matter. These are entirely new insights provided by our model that could be used to guide which regions are most useful as biomarkers at which stages in the disease; and also provide new insights into the underlying atrophy dynamics, which could inform mechanistic models of disease spread in the brain. We believe this result substantially improves the interpretation of our method and will add it to the paper.
>
> Note that to enable direct comparison with previous published results we would need to run the same segmentation tool that they used, e.g., the GIF segmentation tool used in Wijeratne et al. 2023 and Wijeratne & Alexander 2021. This would make an interesting additional analysis that we reserve for future work.
>
> Weaknesses point 2 (“Overall, this work gives the audience…”)
>
> It is a fair point that a key component of the methodological improvement is the application of the Sinkhorn algorithm. However, the reframing of latent variable disease progression models as an optimal transport problem is entirely novel, and has implications beyond just improving the speed of inference (which is what we focused on this paper). Indeed one of the other Reviewers (Vfdn) notes this as one of the paper's strengths: "It is quite innovative to view the disease progression modeling task from the optimal transport perspective.".
>
> The key innovation from an optimisation perspective is moving from a discrete sequence of events to a continuous probability density matrix over permutations – it is this that allows us to unlock gradient-based optimisation via variational inference. Furthermore - and which we don’t explore in depth in the main paper but do provide some initial analysis of in the supplementary - this formulation gives us direct access to the distribution over event permutations (as opposed to having to sample it using MCMC). This allows for direct sampling of model uncertainty in our Bayesian formulation.
>
> But on a more fundamental perspective, one can think of modelling disease as morphing healthy probability distributions into unhealthy ones, (e.g., the diffusion of pathogens spreading in the brain), which can be ideally solved using the tools of optimal transport. For example, one could imagine many types of state-space-based latent variable models that would work with our permutation-based optimal transport coupling, e.g., Hidden Markov models. We’re excited about exploring the many recent developments in optimal transport theory from this perspective in the future.
>
> Weaknesses point 3 (“Practicality of method in real life problems…”)
>
> We need to point out that this criticism is not quite fair – while the simulations show 200 features (a decision we made to facilitate visualisation in the positional variance diagrams shown in Figure 3), the ADNI analysis used 1344 features (as noted in the Figure 4 caption). In principle we could include 10s of thousands of pixels / voxels, the main limitation being identifiability and computer memory, as noted in the Limitations section. However we appreciate that this detail is not easy to find in the text, so we have moved the information about the number of pixel features used in each analysis to the main text in Sections 3.3 and 3.4.
>
> Questions (“In both figure 4, and figure 6, …”)
>
> Yes this is not very clear – we will clarify by replacing these sentences with: “White pixels correspond to events that have occurred by the corresponding point of the sequence.”.

---

> > ### Comment · Reviewer_5r2K · 2024-08-07
> > **Additional question**
> >
> > Thanks for the additional results. I am a bit confused by the definition of the y-axis,
> >
> > > "The horizontal axis shows the event number (from 0 – 1344), and the vertical axis shows the fraction of pixel-events that have occurred in each regional brain volume at the corresponding event number."
> >
> > and why as event number goes up, the fraction also becomes higher?

---

> ### Author Response · Authors · 2024-08-08
>
> Thanks for the question. The "Fraction occurred" label corresponds to the fraction of pixels in each region that have become abnormal, as defined by the vEBM event sequence. The fraction of pixel-events occurred increases with the event number because the event sequence represents a monotonic accumulation in pixel-level abnormality. For example, if we look at the "Cerebral-Cortex" line, it shows the fraction of pixel-events within that region that have occurred as a function of their position in the event sequence, e.g., at "Event" = 600, which is nearing the half way point in the event sequence, approximately 60% of the pixel-events in "Cerebral-Cortex" have occurred.

---

> > ### Comment · Reviewer_5r2K · 2024-08-10
> >
> > Thanks for the clarifications. To double confirm my understanding, event number is basically timesteps, and fraction is the number of abnormal pixels over the total number of pixels in that region. Presumably, the disease progresses it should become higher. That is fair.
> >
> > However, my comment about the quantitative measurement against the ground truth trajectory refers to something else. Since the algorithm is predicting pixel level label, at a specific event/timestep,  Intersection over Union should be computed to measure how realistic the progression simulation is, given a ground truth trajectory.

---

> > > ### Author Response · Authors · 2024-08-12
> > >
> > > Yes your understanding is exactly correct.
> > >
> > > Ok we see your point - comparing to a pixel-level, i.e., correlated feature, ground truth trajectory would indeed be instructive, and given the time, we would have set up a suitable simulation to test this. As it was, we only had time to validate the method using ground truth trajectories of uncorrelated features (Sections 3.2.2 and Supplementary A.7). However, as you can see from our results, our method performs very well at recovering the ground truth trajectory for various dataset properties and model parameter settings, and generally outperforms the two baselines. As you suggest, implementing pixel-level simulations are a priority for our future work.
> > >
> > > On a related note - it would be very difficult to validate the results from the real data analyses in Alzheimer's disease and AMD, due to the standard problem in medical imaging of having no ground truth. The closest one could get is to have an additional dataset of post mortem histology images, matched to the in vivo medical images that we used here, in order to validate the spatial distribution of abnormality predicted by the model. Indeed there are approaches developed by other researchers to register medical images with histology, but they are still experimental and would require substantial time and cross-collaborative effort to implement. That being said, it would be a very interesting direction to take our model, as it provides the first pixel-level predictions of its type, which would support more direct comparison between MRI and histology.

---

> ### Comment · Area_Chair_2ewa · 2024-08-12
>
> Hello reviewer 5r2K,
>
> Thanks for already engaging in discussion with the authors.
>
> I just wanted to check whether the most recent author response has adequately addressed your concerns. Separately, please indicate whether you're sticking with your original score, or if this discussion has led to any change in score on your end. Note that the author/reviewer discussion period ends very soon (Aug 13, 11:59pm AoE).
>
> Thanks,
> Your AC

---

> > ### Comment · Reviewer_5r2K · 2024-08-12
> >
> > Thanks for the responses. My concerns are mostly addressed and looking forward to your follow-up works.

---

### Official Review · Reviewer_Vfdn · 2024-07-04

**Soundness:** 3
**Presentation:** 2
**Contribution:** 3
**Rating:** 6
**Confidence:** 3

**Summary:**

The authors investigate the task of disease progression modeling, an area of research that learns underlying disease trajectory from temporal snapshots of individual patients. The authors claim that all previous approaches either sacrifice computational tractability for direct interpretability in the feature space or vice versa. The authors introduce the variational event-based model (vEBM) to remedy the former situation, by enabling high dimensional interpretable models through a computationally efficient approach that circumvents dimensionality reduction or manual feature selection. vEBM borrows concepts from optimal transport to directly infer a continuous probability over events. The authors further claimed a 1000x speed-up, better accuracy and improved robustness to noise.

**Strengths:**

1. It is quite innovative to view the disease progression modeling task from the optimal transport perspective.
2. Figure 1 makes the paper slightly easier to read, as it outlines the proposed method with references to subsequent paper sections.
3. The proposed vEBM, compared to the baselines (EBM and ALPACA), show significant advantages in efficiency evaluated by wall-clock time. vEBM also shows better scaling with data dimensionality.
4. The datasets the authors used for empirical results (Alzheimer’s disease and age-related macular degeneration) are of significant clinical importance. I particularly like that the authors compare the disease progression patterns with the known changes from the literature.

**Weaknesses:**

1. Correct me if I am wrong: the proposed method seems to be a method that models the disease progression on a population level. This might limit the method to population level studies for disease research purposes and render it unsuitable for predicting individual-level progressions which could facilitate personalized treatment plans.
2. While producing pixel-level disease progression sequences for certain diseases is fantastic, I would suspect the proposed vEBM method is not ideal for pixel-level predictions, since vEBM presumably treats different pixels as separate features, ignoring the spatial information formed by multiple nearby pixels.
3. For the results in Figure 7, while it is visually informative, it will be great if the authors can incorporate quantitative metrics.
4. Minor issue: For Figure 1, it would be great if the authors can improve the aesthetics.
5. Minor issue: For Figure 5, it would be helpful to provide the color bar.

**Questions:**

1. May I ask the authors to explain in a little more detail what Figure 3 Bottom row is visualizing? I am unfamiliar with the positional variance diagrams and it will be helpful if the authors can explain what the ordering of the feature (vertical axis) indicates, what specific traits on the diagram tells us, etc.
2. Would the authors consider comparing with alternative deep-learning-based models, such as Transformers, neural ODE, latent ODE aka ODE-RNN, neural CDE, etc.? If not, could they provide justifications?

**Limitations:**

Yes, the authors adequately addressed the limitations and, if applicable, potential negative societal impact of their work.

---

> ### Author Rebuttal · Authors · 2024-08-06
>
> Weaknesses point 1 (“Correct me if I am wrong…”)
>
> Yes, that is exactly what our model does. It is less of a device for individual level predictions (although can be used that way; see Figure 7, where we use it to estimate individual-level stages along the group-level sequence) and more to elucidate group level patterns of change that aid disease understanding, biomarker discovery, structure heterogeneity, and inspire new treatment development. As we note in the Introduction (L25-29), this class of technique has proved very influential over the last 10 years; in the paper we reference 15 examples of applications in various diseases, and 4 examples of direct methodological developments on the original model, but there are many more applications and methods developments that have been published.
>
> Weaknesses point 2 (“While producing pixel-level disease progression sequences…”)
>
> Yes that's true. A separate grouping procedure would be beneficial and something we conserve for future work here now we have the computational framework in place. As noted in Limitations (L295-298), a first approach would follow reference [35] and use a Markov random field to impose local structure.
>
> Weaknesses point 3 (“For the results in Figure 7…”)
>
> The main point of these plots is to show the individual-level utility of the method, in terms of the distributions of stages for each group. But we are happy to discuss ideas to make this more quantitative; does the Reviewer have anything specific in mind?
>
> Weaknesses points 4 & 5
>
> We have deliberately made Figure 1 greyscale for printing purposes & accessibility, if that is what the Reviewer is suggesting regarding aesthetics. We will add a color bar to Figure 5.
>
> Questions 1 (“May I ask the authors to explain…”)
>
> The positional variance diagram shows the most likely ordering of events (horizontal axis), as characterised by changes in the features (vertical axis). The model also estimates the uncertainty in the positioning of the events, shown by the shaded heatmap in the Supplementary Figures 11-13. Typically these diagrams are used as the main visualisation of event-based model sequences, but due to the large numbers of features our method enables, they become sub-optimal for visualisation purposes. However the main purpose of including them in Figure 3 was to visually demonstrate the agreement between the vEBM inferred sequence and the true sequence, using synthetic data. In real data, e.g., the ADNI experiments, it makes more sense to visualise the results in pixel space, to preserve the spatial information more clearly.
>
> Questions 2 (“Would the authors consider comparing with alternative deep-learning-based models…”)
>
> These model classes make sense for making predictions, but they lack the interpretability of the EBM class of model, which is a key output. Additionally, they typically require longitudinal data, while the EBM class models operate using only cross-sectional data; thus limiting ODE-type models to using only cross-sectional data would be an unfair comparison.

---

> > ### Comment · Reviewer_Vfdn · 2024-08-11
> > **Response to rebuttal**
> >
> > Many thanks to the authors for the rebuttal. I like the general clarifications. Answering a few questions from the authors:
> >
> > 1. For quantitative metrics on Figure 7, I do not have specific metrics in mind. However,  since you are analyzing distributions, would it be helpful to consider divergence measures (KL or JS) or earth mover distance?
> >
> > 2. Regarding the aesthetics, I have no issue with black and white figures. I was speaking of (1) the method figure is not quite beautiful and engaging, but rather looks a bit dull and almost like a casual sketch, and (2) some line plots are using the default color (blue, orange, green), non-obvious error bars, etc., that are not very optimized.

---

> ### Author Response · Authors · 2024-08-11
>
> Thanks for the clarifications.
>
> Before we respond to points 1 and 2, we'd like to ask why you decided to downgrade your review from a 5 to a 4? Your comment seems to be positive so we're unsure as to what we did to cause the score to be downgraded - some detail would be very helpful, so we can try to respond as best as possible.

---

> > ### Comment · Reviewer_Vfdn · 2024-08-11
> > **Thanks for reminder**
> >
> > Thanks for the authors for reminding that. I meant to update the rating from 5 to 6. Misclick from phone. Corrected.

---

> > > ### Author Response · Authors · 2024-08-12
> > >
> > > No problem! Thanks for the positive response.
> > >
> > > In response to your points above:
> > >
> > > 1. Comparing the distributions of stages between clinical labels using a suitable distance metric is a nice idea - we'll add that to the text.
> > > 2. Ok we understand the criticism - indeed the figure does not look very professional! We will try to improve the style to make it more visually appealing. Regarding the plots, we need to point out that the colour choice was made to conform to accessibility requirements (the default colours that we use were designed by the authors of matplotlib to be accessible for people who are colour-blind). However we agree that the error bars are not very clear and will make them larger.

---

> > > > ### Comment · Reviewer_Vfdn · 2024-08-13
> > > > **Thanks to authors' efforts**
> > > >
> > > > Thanks to the authors for the reply. Understood and best wishes.

---

### Official Review · Reviewer_SCze · 2024-07-11

**Soundness:** 3
**Presentation:** 3
**Contribution:** 3
**Rating:** 7
**Confidence:** 4

**Summary:**

This manuscript derives an Event-Based Model via variational inference and optimal transform. This approach significantly enhances computational efficiency, robustness to noise, and scalability, outperforming current methods by a factor of 1000 in wall-clock.

**Strengths:**

1. The experiments have been performed with multiple datasets: synthetic data, neuroimaging, and optical coherence tomography.
2. The proposed method has been compared to two baselines: the event-based model (EBM) and the Alzheimer’s Disease Probabilistic Cascades (ALPACA) model - on synthetic data.
3. The methodology development was pretty clear and supported by relevant literature.
4. Novel formulation of Event-Based Model via variational inference and optimal transport.

**Weaknesses:**

1. I am not sure about the results in Section 3.3.2. You see that CDR58, MMSE, and RAVLT are pretty far in the Event order. If you check the Temporal Event-Based Model (TEBM) (Wijeratne et al., 2021; Wijeratne et al., 2023), these cognitive test scores can be found earlier in the disease timeline. However, there are differences; they used T1 images, and in your case, you used TBM images. Before making new claims, it will be essential to consider experiments with a previously explored set of features. Importantly, the proposed solution is supposed to solve the scaling question. Hence, it should work with the previous set of features.
2. The authors enable models that express progression at the pixel level compared to the previous region level. However, how the insights will compare to the region-level progression needs to be explored. I suggest exploring it via post hoc analysis by combining pixels back into regions and comparing the ordering.  It needs to be clarified whether pixel-level estimation leads us to a new insight into progression. Because it might be great from a computer science perspective but not from a clinical. Furthermore, measurements based on single pixels are highly susceptible to noise. In addition, pixels are not actually pixels but features because the ADNI images were standardized to the template.  Hence, it is more like the most fine-grained "atlas" to the template.

Wijeratne, Peter A., Daniel C. Alexander, and Alzheimer’s Disease Neuroimaging Initiative. "Learning transition times in event sequences: The temporal event-based model of disease progression." International Conference on Information Processing in Medical Imaging. Cham: Springer International Publishing, 2021.

Wijeratne, Peter A., et al. "The temporal event-based model: Learning event timelines in progressive diseases." Imaging Neuroscience 1 (2023): 1-19.

**Questions:**

My questions were combined with weaknesses.

**Limitations:**

The authors adequately addressed the limitations.

---

> ### Author Rebuttal · Authors · 2024-08-06
>
> Weaknesses point 1 (“I am not sure about the results in Section 3.3.2…”)
>
> Generally we expect cognitive test scores to appear later than structural changes in MRI - cognitive deficit is a consequence of loss of brain tissue. As stated in L257-260, the cognitive events occur across the latter 2/3rds of the event sequence; this is broadly consistent with the results in Wijeratne et al. 2023 and Wijeratne & Alexander 2021.
> There are differences to the literature (e.g., RAVLT occurs late in our sequence, while in Wijeratne & Alexander 2021 it occurs mid-sequence). However, the cohorts used are different – we use a subset of ADNI individuals with TBM data, while e.g., Wijeratne & Alexander 2021 use a different subset, and Wijeratne et al. 2023 use the entire ADNI cohort. Inter-group heterogeneity in Alzheimer’s disease is well reported (e.g., reference [5]) and differences in the ordering of changes are expected between different subsets of the ADNI cohort. This would make for a very interesting future research direction for our model, to investigate image-based heterogeneity at the pixel-level - instead of just regional level that previous analyses have been limited to.
>
> Weaknesses point 2 (“The authors enable models that express progression at the pixel level compared to the previous region level…”)
>
> Including comparison to regional volumes is an interesting idea, and one which we would have included given the time. We have now included an additional analysis in the “Author Rebuttal” PDF, where we used the FreeSurfer segmentation tool to obtain pixel-level anatomical labels, which we mapped to our “pixel” events to enable interpretation via a subset of regional volumes  (Figure 1 in the uploaded PDF).
>
> Indeed what we see is broadly what we’d expect from previous results – sub-cortical changes (Thalamus-Proper, Putamen, Hippocampus) are earliest, followed by cortical (Cerebral-Cortex) and white matter (Cerebral-White-Matter), and finally ventricular change (Lateral-Ventricle, VentralDC). However our model provides much more fine-grained insights – we now obtain continuous trajectories of change, which capture interesting non-linearities, e.g., in the Thalamus-Proper, Brain-Stem, and Lateral-Ventricle; this contrasts with the more linear changes in the Hippocampus, Cerebral-Cortex, and Cerebral-White-Matter. These are entirely new insights provided by the model that could be used to guide which regions are most useful as biomarkers at which stages in the disease; and also provide new insights into the underlying atrophy dynamics, which could inform mechanistic models of disease spread in the brain. We believe this result substantially improves the interpretation of our method and will add it to the paper.
>
> Note that to enable direct comparison with previous published results we would need to run the same segmentation tool that they used, e.g., the GIF segmentation tool used in Wijeratne et al. 2023 and Wijeratne & Alexander 2021. This would make an interesting additional analysis that we reserve for future work.

---

> > ### Comment · Reviewer_SCze · 2024-08-12
> > **Response to rebuttal**
> >
> > I want to thank you authors for their rebuttal. I decided to increase the score to Accept.

---

> ### Comment · Area_Chair_2ewa · 2024-08-12
>
> Hello reviewer SCze,
>
> Does the author respond adequately address your concerns?
>
> Note that the author/reviewer discussion period ends very soon (Aug 13, 11:59pm AoE).
>
> Thanks,
> Your AC

---

### Official Review · Reviewer_wy4q · 2024-07-12

**Soundness:** 3
**Presentation:** 2
**Contribution:** 2
**Rating:** 7
**Confidence:** 3

**Summary:**

The authors propose a method to learn a latent event sequence from cross-sectional data of modest size. Each event corresponds to a single observed feature transitioning from an initial parametric distribution (i.e., 'normal') to a second, final parametric distribution (i.e., 'abnormal'). Their inference procedure incorporates elements of variational inference and optimal transport. They demonstrate that it outperforms two available baselines both in speed / scalability and accuracy of the learned event sequence. Finally, they apply the method to learn the order in which (a) individual pixels of (carefully registered) MRI images become abnormal in Alzheimer's disease, and (b) individual OCT pixels become abnormal in macular degeneration.

**Strengths:**

Results clearly demonstrate that their method is scalable, and it's effective in simulation where the true data match their model. The writing style is clear, and the visualizations are excellent, particularly Figures 4 and 6. Experimental results are interesting and clearly presented.

**Weaknesses:**

- The related work section is very brief, which made me question whether it is comprehensive.
- I found the second half of section 2.2 very difficult to follow. Specifically, I don't understand the relationship between S and X (see lines 138-143) and think this relationship and the optimal transport details more broadly could be presented much more clearly.
- I am not convinced that this is solving a real problem. The approach learns the order in which pixel-sized image regions tend to become abnormal in AD and AMD, but I would think that the more interesting questions have to do with individual variability in the order and timing of this sequence. I may be wrong about this, but I think the authors should do more to explain how the resulting model might be useful in practice.

**Questions:**

- I am confused about how the normal and abnormal distributions are defined in the synthetic data, and whether they are learned versus fixed during inference. Equation (3) implies to me that they are learned, but other descriptions seem to imply that they are initially learned by dividing the population into patients and controls, but then fixed during the broader inference procedure.
- What is denoted by the color of the pixels in Figure 5?
- The authors mention that the method requires image registration, but very few details are given. It seems to me that it would be challenging to align images from very different stages of progression. Is the method sensitive to this image registration step?
- Do the authors envision applications of this method outside of medicine?

**Limitations:**

Somewhat, but a more comprehensive related work section is needed to understand pros and cons of this method (and the approach to disease progression modeling more generally; see Weaknesses) relative to alternatives.

---

> ### Author Rebuttal · Authors · 2024-08-06
>
> Weaknesses point 1 (“The related work section is very brief…”)
>
> This section was deliberately kept brief to focus on the most relevant comparable methods; specifically those that can infer group-level sequences from cross-sectional data. While there are many models that use longitudinal data (see, e.g., reference [19] for a comprehensive review), they are not as relevant for direct comparison. We also note that we do discuss the relative pros and cons of alternative types of models (including longitudinal models) in the Introduction (L30-36). However we appreciate that this is not made completely clear in the Related Work section, so we will add a couple of sentences to clarify this point.
>
> Weaknesses point 2 (“I found the second half of section 2.2 very difficult to follow…”)
>
> The transport cost matrix, X, defines the likelihood of an event occurring and the permutation matrix, S, defines the event ordering; we seek to find the optimal event permutation that maximises the overall model likelihood. As such we can think of the relationship between S and X as their (element-wise) product giving the likelihood of a given event permutation. Alternatively, we can think of relationship as S being the transport plan that permutes event likelihoods in X to their optimal position in the latent event sequence. We will add clarification to the text, if the Reviewer believes these descriptions are helpful.
>
> Weaknesses point 3 (“I am not convinced this is solving a real problem...”)
>
> Here we use pixel-level regions just as a demonstration of the model’s ability to handle high dimensional data. However the model also enables, e.g., fine-grained regional brain volume analysis of the event ordering with 100s of regions, which current methods cannot cope with.
>
> The class of models we focus on here are necessarily group level models rather than models of individuals, although they do capture variability over the group (but we do not leverage this information here). As we highlight in the Introduction (L25-29), these models have proved extremely important in understanding the temporal evolution of chronic disease and guiding treatment development. They are not just, or even primarily, a device for prediction at the individual level. This class of technique has proved very influential over the last 10 years; in the paper we reference 15 examples of applications in various diseases, and 4 examples of direct methodological developments on the original model, but there are many more applications and methods developments that have been published. The method we provide in our paper here will "turbocharge" all the current models of this class, including the highly popular SuStaIn model (reference [5]), and unlock pixel / voxel scale analyses that were previously intractable.
>
> Questions point 1 (“I am confused about how the normal and abnormal distributions…”)
>
> These distributions are defined by separating synthetically generated individuals into two groups, based on their randomly generated stage (the lowest 20% of stages are considered controls, see L194-195). Mixture models are then fitted to these groups, and are fixed throughout inference, as denoted in Figure 1. Equation 3 doesn’t explicitly distinguish between these fitted parameters (“\theta”) and the learned parameter “S”, because in principle they could be learned jointly; however if the Reviewer thinks it is helpful, we can clarify the difference by using semicolon notation for "\theta" in the likelihood equations.
>
> Questions point 2 (“What is denoted by the color of the pixels in Figure 5?”)
>
> The color denotes the number of pixel events in each histogram bin, e.g., in the first bin of events (the first column), we can see the density of pixel events occurring as a function of the distance from the centre. We will add a color bar and a sentence in the figure caption to clarify this.
>
> Questions point 3 (“The authors mention that the method requires image registration…”)
>
> We deliberately avoided going into detail regarding the registration, because a) we used a pre-processed data collection, produced using Tensor Based Morphometry, from the ADNI dataset, which has previously been described in detail (see reference [59]); and b) as we note in the Limitations (L292-295), some sort of pre-processing is always necessary for any method to enable comparison between individuals, so it isn’t a special limitation of our method.
>
> Regarding the point about aligning images from different stages of progression – if we understand correctly, this is actually the primary utility of disease progression modelling, which can be thought of as a type of temporal registration of images generated by latent stages to a common temporal reference frame. Our method captures this variability and learns this temporal reference frame in terms of a sequence of events. However if the Reviewer is referring to the spatial registration of images with respect to a common spatial reference template, the primary utility of Tensor Based Morphometry is to provide a method to map to a common reference template and calculate the voxel-level volumetric change with respect to this template, thus transforming all images from different stages to a common spatial reference frame.
>
> Questions point 4 (“Do the authors envisage applications of this method outside medicine?”)
>
> Yes, we envisage many applications in areas that involve learning progressive sequences of events, e.g., in environmental modelling, our method can infer trajectories of biodiversity loss “events”, from temporal eco-acoustic monitoring data. We plan to make the vEBM code available with the aim of opening up such applications by researchers in other fields.

---

> > ### Comment · Reviewer_wy4q · 2024-08-08
> >
> > Thanks for your responses and clarifications, which are very helpful and address my concerns. I do think that including some of these details / clarifications in the text itself would be helpful. Most important are the clarifications you provide in your responses to Weaknesses point 2 and Questions points 1-2. I have increased my score.

---

> > > ### Author Response · Authors · 2024-08-09
> > >
> > > Thanks for the positive response. We will include the clarifications you highlight in the manuscript.

---

### Author Rebuttal · Authors · 2024-08-06

We thank the reviewers for their constructive comments, which we have endeavored to address as faithfully as possible in our responses below.

As part of our response, please find attached a PDF containing a new result.

We look forward to continuing the constructive discussion!

---

### Decision · Program_Chairs · 2024-09-25

**Decision:**

Accept (poster)

**Comment:**

Thanks to the reviewers and the authors for engaging in discussion, and the authors for adequately addressing reviewer concerns. The reviewers all lean toward acceptance, so I will favor acceptance as well.